# ASIF: COUPLED DATA TURNS UNIMODAL MODELS TO MULTIMODAL WITHOUT TRAINING

## ABSTRACT

Aligning the visual and language spaces requires to train deep neural networks from scratch on giant multimodal datasets; CLIP (Radford et al., 2021) trains both an image and a text encoder, while LiT (Zhai et al., 2022) manages to train just the latter by taking advantage of a pretrained vision network. In this paper, we show that sparse relative representations are sufficient to align text and images without training any network. Our method relies on readily available single-domain encoders (trained with or without supervision) and a modest (in comparison) number of image-text pairs. ASIF redefines what constitutes a multimodal model by explicitly disentangling memory from processing: here the model is defined by the embedded pairs of all the entries in the multimodal dataset, in addition to the parameters of the two encoders. Experiments on standard zero-shot visual benchmarks demonstrate the typical transfer ability of image-text models. Overall, our method represents a simple yet surprisingly strong baseline for foundation multimodal models, raising important questions on their data efficiency and on the role of retrieval in machine learning.

## 1 INTRODUCTION

Large multimodal models such as CLIP (Radford et al., 2021) are rapidly becoming the standard for foundation models (Bommasani et al., 2021) in computer vision. This is largely due to their zero-shot and open-world capabilities that enable diverse suites of downstream tasks, from classification to detection and visual search. Overall, Radford et al. (2021) demonstrated that *scale* is the key ingredient for building a common latent space for images and text, and is sufficient to convincingly solve a multitude of tasks without training explicitly for them. In fact, CLIP's zero-shot classification accuracy on Imagenet (Deng et al., 2009) drops from 76.2 to 31.3 when using a public dataset of "just" 15M pairs (a curated subsampled

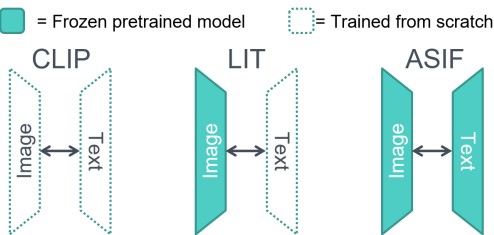

Figure 1: ASIF is a simple recipe to align the representations of two frozen pre-trained models exploiting the fact that relative distances are preserved across different modes: the captions of similar images are themselves similar.

version of YFCC100m (Thomee et al., 2016)) as opposed to the original private dataset of 400M pairs. Recently, Zhai et al. (2022) showed improvements in data efficiency by only training the text encoder, achieving Imagenet accuracy comparable to CLIP with 10M samples and outperforming it with a larger 901M data set.

Training models at such scale presents several challenges beside the obvious infrastructure and training costs. Notably, it requires collecting massive training sets, making it difficult to interpret the predictions of the model in light of their training data. Additionally, the training assets are often not owned by the institution training the model (Sun et al., 2017). This introduces several additional challenges, from reproducibility to the difficulty of ensuring that an asset owner can remove their data from the model (Golatkar et al., 2021; 2020b;a; Ginart et al., 2019; Guo et al., 2020). Overall, these considerations make large multi-modal models relatively inaccessible to researchers and practitioners until checkpoints are released or access to demo is granted.

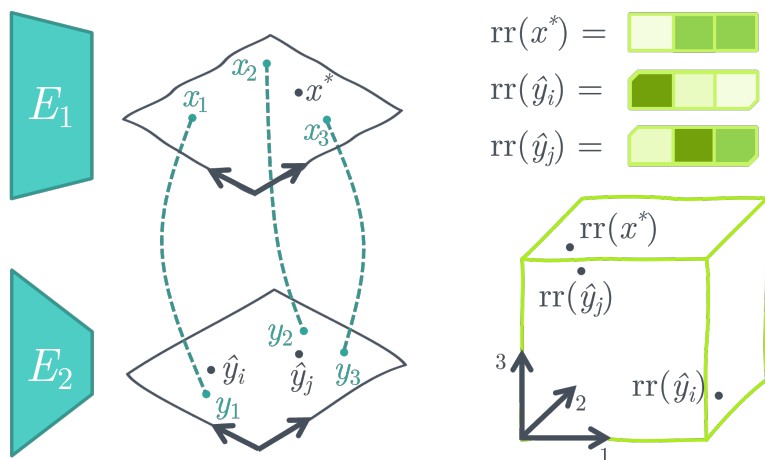

Figure 2: **The ASIF construction.** An ASIF model is defined by two unimodal pretrained encoders and a collection of coupled embeddings (in turquoise). This is sufficient to compare elements from different modes through their relative representations: $rr(\hat{y}_j)$ is more similar to $rr(x^*)$ than $rr(\hat{y}_i)$.

In this paper, we present ASIF, a simple procedure that turns pre-trained uni-modal image and text models into a multi-modal model using a *relatively small*[1] multi-modal data set and no additional training as shown in Figure 1. The resulting model aligns latent representations of images and text, behaving *as if* it was contrastively trained on multi-modal data like CLIP or LiT. The key intuition is that captions of similar images should be themselves similar, and therefore a representation crafted using just similarities to ground-truth multimodal pairs is quasi mode-invariant.

Our results are surprising and raise several questions. Despite (1) the simplicity of the approach, (2) a multi-modal data set that is up to 250 times smaller than in prior work and (3) the lack of actually training the model on multi-modal data, ASIF achieves zero-shot classification accuracy on downstream data sets that is roughly in the same ballpark of CLIP (Radford et al., 2021; Zhai et al., 2022). This raises important questions on the data efficiency in foundation models, making ASIF a very powerful and cheap baseline for future work, and opening new doors for data centric AI (Ng, 2022). In fact, ASIF comes with several interesting properties by construction. The absence of training makes the model editable: adding or removing image-text pairs and deploying a new multimodal model is a matter of seconds. Moreover, the representations are highly interpretable, since every dimension corresponds to the similarity of the input to a unique entry in the multimodal dataset.

In summary, we:

- Introduce the ASIF procedure, which turns two pretrained unimodal black-box encoders into an interpretable multimodal model without tuning a neuron and using a "limited" amount of multimodal data.
- Demonstrate the effectiveness of ASIF models on zero-shot image classification tasks, where they achieve performance in the same ballpark of CLIP with significantly fewer image-text pairs.
- Discuss key properties of ASIF, its implications on the role of memory and retrieval in machine learning, and the new opportunities it opens.

## 2 ALIGNING PRE-TRAINED MODELS VIA RELATIVE REPRESENTATIONS

In the following we present how a collection of captioned pictures implicitly defines a common space for images and texts through relative representations, allowing to build a multimodal model without training. Before that, we briefly discuss existing techniques to build this common space.

---

[1] Compared to the data sets used to train state-of-the-art multimodal models, our experiments use 1.6M captioned images.

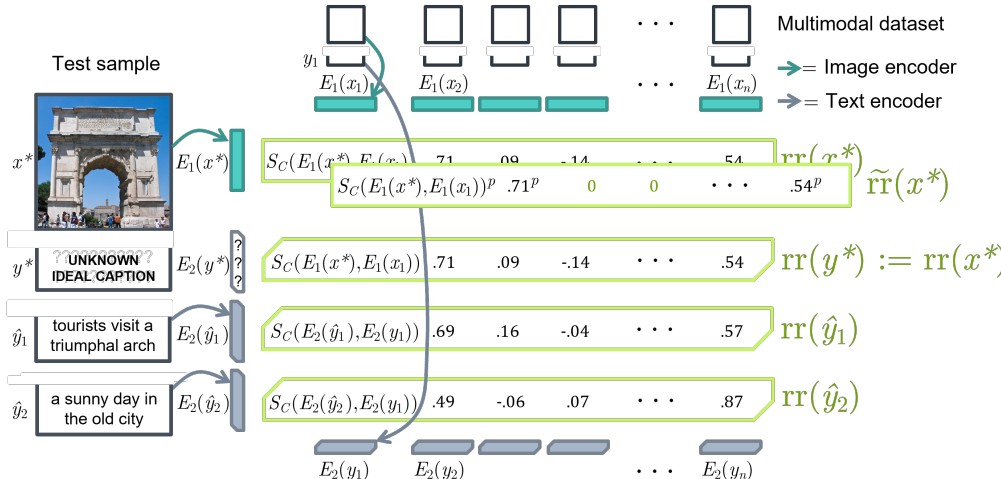

**Figure 3: Zero shot classification.** The best caption is identified by picking the one with its processed relative representation $\tilde{rr}(\hat{y}_i)$ closest to the input image $\tilde{rr}(x^*)$. In other words, we consider the relative representation of $x^*$ with respect to the image collection $x_1, \ldots, x_n$ *as if* it was the relative representation of $y^*$ – the ideal caption for $x^*$ – with respect to the corresponding caption collection $y_1, \ldots, y_n$.

**Contrastive training to build a common space.** With multimodal models, we refer to architectures that embed inputs of diverse modes into the same space. The better is a multimodal model, the closer are representations of different modes of the same object. So far, this common space has been obtained as the result of a contrastive training of one (Zhai et al., 2022) or both the neural mode encoders (Radford et al., 2021; Jia et al., 2021). Using e.g. a collection of image-text pairs as training set, a contrastive loss promotes the closeness of image and text embeddings from the same pair, while spacing out the ones from distinct pairs. Zhai et al. (2022) train just the text encoder to match the image embeddings of a pretrained visual encoder. Once the encoders are trained, a multimodal model can be adapted to perform any visual classification task just by crafting a caption for each label and selecting the one with the embedding closer to the embedding of the input image.

**Relative representations.** Our idea to build a common latent space is to use a collection of coupled data as a "rosetta stone" between modalities, and represent each new data point as its similarities to the points of the same modality in the collection. In other words, we compute a *relative representation* for each new data point:

**Definition 2.1** *Given an encoder $E : X \to \mathbb{R}^d$ and a subset of samples $\{x_1, \ldots, x_n\}$ denoted as anchors, we define the relative representation of $x'$ as the $n$-dimensional vector:*

$$rr(x') = (sim(E(x'), E(x_1)), \ \ldots \ , sim(E(x'), E(x_n)))$$

*for some similarity function sim, such as the cosine similarity.*

Relative representations of points from different modalities live in the same latent space if they are with respect to the same multimodal collection, see Figure 2.

**Relation with Kernel methods.** Definition 2.1 may not look surprising to readers familiar with the literature on kernel methods (Hofmann et al., 2008). Instead of presenting $rr(x')$ as a kernel, we say it is a relative representation to stress that (1) we want to *explicitly* represent the coordinates in our ASIF recipe as opposed to operating in an implicit feature space and (2) we do not aim at learning regression parameters, although we note this may help with the inference speed. Instead, we rely on a simpler procedure that may be interpreted as a hard version of the Watson-Nadaraya (Nadaraya, 1964; Watson, 1964) regression with a distance threshold. Other benefits from incorporating kernel methods explicitly are left to future works, as our work focuses on demonstrating how single-domain pre-trained networks can be aligned without further training.

**ASIF: relative representations inducing a meaningful common space.** Consider the embedding spaces of any two good text and visual encoders[2], we expect captions of images that are close in the visual space to be themselves close in the language space. This fact makes a representation defined in terms of similarities against a set of ground-truth multimodal pairs almost mode-invariant, i.e. an image and its caption share almost the same representation.

That is why we can assign the best caption to a new image $x^*$ just by performing nearest neighbors: we can consider the relative representation of $x^*$ respect to the image collection $(x_1, \ldots, x_n)$ *as if* it was the relative representation of its ideal caption $y^*$ with respect to the counterpart collection $(y_1, \ldots, y_n)$, see Figure 3.

The whole procedure to set up an ASIF model and use it to find the best caption for a new image follows.

---

**ASIF recipe.** Ingredients:

- Two good encoders, each mapping a single data modality to a vector space. Let $X$ and $Y$ be the mode domains, for instance a pixel space and a text space, we need $E_1 : X \to \mathbb{R}^{d1}$ and $E_2 : Y \to \mathbb{R}^{d2}$.

- A collection of ground truth multimodal pairs: $D = \{(x_1, y_1), \ldots, (x_n, y_n)\}$, for instance captioned images.

Procedure to find the best caption among a set of original ones $\hat{Y} = \{\hat{y}_1, \ldots, \hat{y}_c\}$ for a new image $x^*$:

1. Compute and store the embeddings of the multimodal dataset $D$ with the encoders $E_1, E_2$ and discard $D$. Now in memory there should be just $D_E = \{(E_1(x_1), E_2(y_1)), \ldots, (E_1(x_n), E_2(y_n))\}$;

2. Compute the $n$-dimensional relative representation for each candidate caption $\text{rr}(\hat{y}_i) = (\text{sim}(E_2(\hat{y}_i), E_2(y_1)), \ldots, \text{sim}(E_2(\hat{y}_i), E_2(y_n))$, where sim is a similarity function, e.g. cosine similarity. Then for each $\text{rr}(\hat{y}_i)$ set to zero all dimensions except for the highest $k$, and raise them to $p \geq 1$. Finally normalize and store the processed $c$ vectors $\tilde{\text{rr}}(\hat{y}_i)$. Choose $k$ and $p$ to taste, in our experiments $k = 800$ and $p = 8$;

3. Compute the relative representation of $x^*$ using the other half of the embedded multimodal dataset $D_E$ and repeat the same processing with the chosen $k$ and $p$;

4. We consider the relative representation of the new image $x^*$ *as if* it was the relative representation of its ideal caption $y^*$, i.e. we define $\tilde{\text{rr}}(y^*) := \tilde{\text{rr}}(x^*)$. So we choose the candidate caption $\hat{y}_i$ most similar to the ideal one, with $i = \text{argmax}_i(\text{sim}(\tilde{\text{rr}}(y^*), \tilde{\text{rr}}(\hat{y}_i)))$.

To assign one of the captions to a different image $x^{**}$ repeat from step 3.

---

**Properties of ASIF models.** The above construction yields several intriguing properties for free:

*No training and "data centric"*. As we have seen, an ASIF model is built on top of two independently pretrained encoders and the embeddings of a multimodal dataset, and so without training or finetuning any neuron. Being deployable or updatable in seconds, an ASIF model is radically "data centric" (Ng, 2022). For example, it is trivial to adjust the model by adding or forgetting specific samples. The latter use-case is particularly important, as the right to use specific assets may change over time and removing the effect of specific samples from a trained network requires sophisticated forgetting techniques, e.g. Golatkar et al. (2021; 2020b;a); Ginart et al. (2019); Guo et al. (2020). In our procedure, the encoders can be pre-trained with established data sets that do not change over time, and removing the effect of a multi-modal example is as simple as deleting its embeddings.

*Data efficiency:* Being able to exploit two pretrained encoders, ASIF models require far less ground-truth multimodal pairs to become effective. As confirmed by our experiments, ASIF reaches competitive zero-shot performance on diverse classification benchmarks by using a fraction of the mul-

---

[2]Vision and language are the only modalities we target in this paper due to wider availabilty of models and paired data, although we expect the procedure to be more general.

timodal data of its predecessors, reaching a respectable accuracy even with thousands of pairs (we refer to Section 3 for more details). This is in line with classical work in computer vision, where prototypical networks (Snell et al., 2017) are a strong baseline in the extremely few-shot regime.

*Interpretability:* Sparse relative representations make the ASIF models interpretable classifiers. In fact, we can trace back every prediction to a small set of data points in the multimodal dataset – corresponding to the dimensions that are nonzero both in the image and the caption-label– responsible for the outcome (at most $k$), see Figure 5. This enables visualizations of the relevant samples contributing to a correct or incorrect prediction at no extra cost, in stark contrast with other approaches that are often costly (Achille et al., 2021) and fragile (Koh & Liang, 2017; Basu et al., 2021).

**Key limitation – Implementation that scales.** Clearly, our method pays the price of avoiding training with scalability at inference time. As such, our approach should not be considered a general one-stop replacement for CLIP, although in our experiments we could scale ASIF to 1.6M pairs with reasonable inference speed (non-optimized ASIF nearest neighbor procedure made inference speed less than 2x slower). If someone wanted to truly deploy a scalable implementation of ASIF, there are several components that facilitate scalability. Having just $k$ nonzero entries, relative representations can be implemented using sparse vectors, with cosine similarities computed quickly in batches through sparse matrix multiplication. Relative representations can be constructed and compared efficiently even when the multimodal dataset scales up to many millions of entries, thanks to established techniques such as product quantization (Jegou et al., 2010) and inverse indexing (Sivic & Zisserman, 2003), both implemented e.g. in the FAISS library (Johnson et al., 2019). Finally, we find that the distribution of pairs chosen during inference is rather short tailed, presenting opportunities to prune the model even *at deployment time*, deleting from memory the data that is never used. Other more advanced techniques like coresets (Bachem et al., 2017) are also possible.

## 2.1 Design choices and implementation of ASIF models.

**Curating the multimodal dataset.** While neural models like CLIP or LiT are defined just by the weights of the two encoders, to univocally identify an ASIF model we need also the embeddings of the multimodal dataset. Even if two image-text ASIF models rely on the very same encoders, they comply to different visual understandings of the world if they are based on different collections of image-text pairs, therefore achieving different performances on the same tasks.

In contrast to traditional neural vision models, the ASIF approach makes an effective curation of the training dataset possible, since it works even with very small multimodal datasets and allows to evaluate the impact of every data point, as seen in Figure 5.

**Salient hyperparameters.** While using the raw relative representations already provide an ASIF multimodal model with non-trivial capabilities, we found that two simple treatments greatly improve performance, efficiency, and also foster interpretability.

(i) *Sparsification.* We set to 0 all the entries of the $n$-dimensional relative representation except for the top $k$. In our experiments $n$ and $k$ are respectively in the order of millions and hundreds. In this way we cut off the small noisy signals from the dissimilar entries, that accumulated during comparisons would destroy the signal from the few similar entries. Furthermore we get highly interpretable representations that can be efficiently compared, since we have just $k$ nonzero features, each one linked to a single entry in the multimodal dataset.

(ii) *Exponentiation.* We raise all the nonzeroed similarities $\text{sim}(E(x'), E(x_i))$ to $p$, with $p \geq 1$. This non-linearity weighs more the contribution of the most similar entries in the relative representation.

Besides the pivotal choice of the ground-truth multimodal pairs, the number of non-zero elements $k$ and the exponent $p$ are the salient hyperparameters to consider when deploying an ASIF model. In general, we found that picking a $p \neq 1$ may help, while choosing a $k \ll n$ is always crucial. For more details see Section 3.

## 2.2 Closely Related Works

**Retrieval augmented foundation models.** In (Rota, 1985), Stanley Ulam affirms that a mathematical formalization of the word "as" –on a par with the connectives "and", "or", "implies" and "not"– would be a key milestone to artificial intelligence. The idea of having analogies at the core

of cognition is shared by Hofstadter (2001) which states that a concept is a package of analogies, coherently to what prescribes the ASIF procedure. This inductive bias can be implemented in machine learning considering relational information and retrieval augmented models. Recent works in NLP enhance unimodal language models with retrieval to reduce the size of the architectures and the training dataset while also making results more transparent (Borgeaud et al., 2022; Izacard et al., 2022). Our work is in line with this view, that the ASIF procedure extends to the multimodal case. Importantly, ASIF offers a new perspective on data centric AI (Ng, 2022), where data and the external memory implement the alignment between modalities. Networks with discrete Key-Value bottlenecks (Träuble et al., 2022) are closely related to our approach, with the critical differences that our memory is not learned and that their decoder is trained for classification. Key-Value bottlenecks as in (Träuble et al., 2022) could be good candidates to implement a summarization of the data set that is learned and of fixed size. Finally, we notice that making predictions on new samples by exploiting the similarity with a dictionary of previous data points is a very common approach in computer vision (Snell et al., 2017) for few-shot learning. Our procedure is also related to compressed sensing algorithms where a signal (in our case an image) is sensed as a sparse combination of fixed atoms (Wang et al., 2012; Mallat & Zhang, 1993) with an iterative projection procedure (Locatello et al., 2017; 2018) and only transmitting the coefficients to the receiver (text modality).

**Learning multimodal models.** Following the intuition outlined by early works on aligning text and image embeddings (Frome et al., 2013; Karpathy & Fei-Fei, 2015), today large multimodal models are conquering the computer vision scene thanks to their wide applicability and easy transfer to new downstream tasks (Radford et al., 2021; Zhai et al., 2022; Jia et al., 2021; Yu et al., 2022). We identify two key leaps respect to traditional models like ResNet (He et al., 2016): (i) Free text allows to learn visual concepts beyond a finite set of predefined categories and to exploit the structure of language to compose them, as masterfully seen in Dall-E (Ramesh et al., 2022). (ii) The recognition tag transitioned from being an output pulled out of the image by the neural stack (the label) to become an input that should be interpreted, and therefore processed by its own encoder (the free text caption). This corresponds to an epistemological perspective shift, as discussed by Norelli et al. (2022).

Data and learning efficiency are clear challenges for large multimodal models, that often require hundreds of millions of examples. Efforts such as (Zhai et al., 2022; Izacard et al., 2022) attempt to reduce this. ASIF presents a different take on this problem, showing how much can be achieved by simply remembering the training data efficiently.

## 3 EMPIRICAL EVIDENCE

In the following we compare ASIF to traditional multimodal models based on contrastive training, CLIP and LIT. We then take a closer look to the classification of a single image, unpacking the relative representations and following the classification algorithm step by step. As a prelude to the above, we start by discussing the pretrained encoders and dataset forming the ASIF models we tested.

**Pretrained encoders and multimodal dataset used.** In our experiments we used visual transformers as image encoders, either trained in a supervised (VITb16, Dosovitskiy et al., 2021) or unsupervised way (DINO VITs8, Caron et al., 2021). Both transformers were pretrained on Imagenet 21k (Ridnik et al., 2021). In the supervised case, the labels were used during training, while in the unsupervised case, the model learned to represent two views of the same image in the same way. The embedding size is 768 for VITb16 and 384 for DINO VITs8. Note that this is the same setup as in Zhai et al. (2022). The only difference with Zhai et al. (2022) is that they train the text encoder and we use a frozen pre-trained one. Concerning the text encoder, we used the SentenceT pretrained transformer (Reimers & Gurevych, 2019). Its training dataset consists of more than 1 billion sentences scraped from the internet. Here there are multiple learning tasks used for pretraining, including a contrastive one with sentence pairs. The embedding size of SentenceT is 768.

As multimodal dataset we used the first 1.6M entries of the Conceptual Caption dataset (CC12M, Changpinyo et al., 2021), which collects images and filtered alt-texts scraped from the internet.

**Zero-shot performance.** We assessed the quality of our ASIF multimodal model by comparing its zero-shot classification performance against CLIP and LIT on four datasets: CIFAR100, Imagenet, Imagenet v2, and PETS; see Table 1. We crafted label prompts as in LIT (Zhai et al., 2022, Table 11).

| Method | Dataset size | ImageNet | CIFAR100 | Pets | ImageNet v2 |
|---|---|---|---|---|---|
| CLIP (Radford et al., 2021) | 400M (private) | 68.6 | 68.7 | 88.9 | - |
| CLIP (Radford et al., 2021) | 15M (public) | 31.3 | - | - | - |
| LIT (Zhai et al., 2022) | 10M (public) | 66.9 | - | - | - |
| CLIP (Zhai et al., 2022, uu) | 901M (private) | 50.6 | 47.9 | 70.3 | 43.3 |
| LIT (Zhai et al., 2022) | 901M (private) | 70.1 | 70.9 | 88.1 | 61.7 |
| ASIF (sup vis. encoder) | 1.6M (public) | 55.4* | 63.3 | 71.5 | 45.6 |
| ASIF (unsup vis. encoder) | 1.6M (public) | 53.0* | 46.5 | 74.7 | 45.9 |

Table 1: **Zero shot classification accuracy of different multimodal designs.** CLIP and LIT implementations vary by dataset and the visual transformer used as image encoder. The first CLIP and LIT entries use a VITb16 as ASIF, the last CLIP and LIT entries use a VITb32 (larger patch size). The public dataset of CLIP is a curated subset of YFCC100m (Thomee et al., 2016), while LIT and ASIF use CC12M. *We used the ImageNet validation set to tune the two hyperparameters of ASIF which were then used on the other data sets. The number reported in the table is a test set.

Remarkably, we achieve competitive performance with CLIP and LIT using two frozen pretrained encoders and a fraction of the image-text pairs.

In Figure 4 we show the full zero-shot accuracy trend on Imagenet as the size of the multimodal dataset grows for different choices of $k$ and $p$. ASIF models become effective very quickly: we reach a non-trivial 18% classification accuracy using just the first 10,000 pairs in the multimodal dataset. We recall that building an ASIF model from scratch still requires a lot of unimodal data to pretrain the encoders, but now this data may come untied and even without any label.

In Table 1 we also report the zero-shot accuracy of two ASIF models on four standard datasets, differing just for the vision encoder, that is respectively supervisedly and unsupervisedly pretrained (DINO). We tuned $k$ and $p$ on the ImageNet validation set, in both cases we used $k = 800$ and $p = 8$.

The comparison between the two versions is not completely fair since the visual transformer architecture of DINO is smaller (e.g. the codings are 384-dimensional rather than 768) but corroborates the effectiveness of ASIF with encoders pretrained using no supervision.

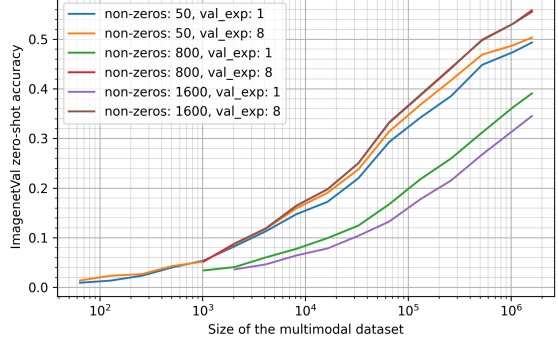

Figure 4: Impact of $k$ (non-zeros), $p$ (val exp), and dataset size on ImageNet zero-shot classification.

*Summary:* Overall, we observe that our ASIF procedure can achieve competitive zero-shot results with a fraction of the image-text pairs used in prior work (assuming access to other pre-trained, even unsupervisedly, uni-modal models).

**Deep dive into a classification.** To better understand why and how the ASIF procedure works, we are going to follow step by step the single classification process depicted in Figure 5 that shows the entries of the multimodal dataset used to build the relative representations. For simplicity we assume $k = 23$.

We want to find the imagenet label of the upper left image in Figure 5. The first step is to compute its relative representation with respect to the multimodal dataset, this procedure selects the 23 most similar images in our collection. The most similar are indeed triumphal archs that –as similarity decreases– turn to generic structures with archs or ancient monuments. No other image in the multimodal dataset is involved in the classification, we could replace all the entries in our collection except for these 23, and as long as no new image is more similar to the input image, every computation in the classification process would remain the same. This consistency of the ASIF procedure is in contrast to traditional approaches like CLIP and LIT, where the parameters defining the encoder computations are shaped by the whole multimodal dataset through contrastive learning.

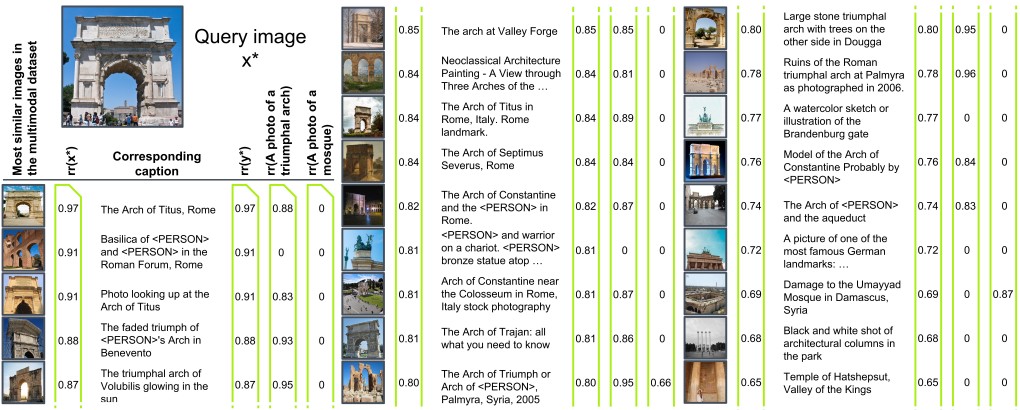

Figure 5: **Deep dive into a classification.** We show step-by-step how ASIF produces predictions for a query image in an interpretable way. By construction, ASIF allows to visualize the samples leading to correct or incorrect predictions which is useful to curate better data.

Now is time to perform the core step of the ASIF procedure: jumping in the text space by considering the relative representation of the input image *as if* it was the relative representation of its ideal caption. So we find ourselves with the same vector, but now meaning how much the ideal caption should be similar to 23 captions. Here the quality of the multimodal dataset matters; looking at the 23 captions we are confident that the candidate caption closer to match the 23 similarities is one of the prompts associated to the `triumphal_arch` label in ImageNet, like *"a photo of a triumphal arch"*. Indeed, a bad multimodal dataset spawning 23 non-informative captions like filenames *"IMG_20180823.jpg"* or camera settings *"D90 18.0-70.0 mm f/3.5-4.5"* would have completely failed in recognizing the correct class, even if supported by the best image and text encoders. Such a catastrophic failure is a feature of the ASIF procedure, since it follows that the sole multimodal dataset give us full control of the model: the two encoders may have seen anything, what matters is the meaning attributed to it by the multimodal dataset.

*Summary:* Our simple (non-cherry picked) example showcases how the ASIF predictions can be easily attributed to specific examples in the training data by construction. This feedback can be used to explain predictions and grow high quality datasets in data centric AI, for example by inspecting which examples contribute to incorrect classifications.

# 4 DISCUSSION

The effectiveness of the ASIF procedure raises questions on the role of memory and retrieval in machine learning, while at the same time opens new opportunities for products based on multimodal models, opening many avenues for future works. In the following we will discuss these aspects.

**Perception and interpretation disentangled.** ASIF redefines what constitutes a multimodal model by explicitly disentangling memory from encoding: here there is no trace of the multimodal data in the weights of the encoders, which are pretrained on different datasets. Nonetheless, the relative representations and the outcome of any classification task fundamentally depend on the multimodal dataset. This state of affairs reflects the factorization of perception and interpretation in the two stages constituting an ASIF model; the encoding and the construction of the relative representations. Such factorization may be desirable, since it relieves the neural encoders from the responsibility of attributing meaning to their inputs. Therefore allowing us to "keep our nose out of the black boxes" (Eco, 2000, Par. 3.3.1.3) and only consider the second stage to analyze, explain, and act on the interpretations of an ASIF model. Similarly to when –to teach the interpretations at the basis of a discipline– we modify the schoolbooks rather than directly adjusting the synapses of the learner.

**Learning or retrieval?** As we have seen, the ASIF procedure requires no training: it does not distill the multimodal data into any learnable parameter. Rather, it prescribes a rigid memorization of the

multimodal dataset, where each entry has its fixed-size spot, similarly to a retrieval process. On the other hand it seems impossible to not describe ASIF as a learning algorithm; for instance it satisfies the fundamental characterization given by Mitchell (1997): the more the multimodal data the more ASIF improves, as we can clearly see e.g. in Figure 4. Ultimately, ASIF is functionally comparable to CLIP. ASIF blurs the border between learning and retrieval by questioning the effectiveness of storing information only in the weights and rather advocates to combine learning representations with external memories. We encourage more work on memory augmented neural networks and towards understanding the implications of memory for generalization.

**Generalization to new distributions.** The empirical performance of ASIF calls for a discussion on zero-shot and out-of-distribution generalization in foundation models trained with huge data sets. Clearly, the performance of ASIF will depend strongly on the multi-modal data used for alignment. As an example, the good performance on Imagenet may not be particularly surprising in light of the qualitative evaluation seen in Figure 5. There, our query image might as well had been part of our multi-modal data set, as the semantic gap with its neighbours appears to be small. Despite this, our choice of downstream evaluation and pre-training data set is identical compared to prior work (Zhai et al., 2022). As such, while it appears clear that ASIF should fail when the semantic gap between downstream task and "training data" is large, it is unclear why it should be different for more standard models like CLIP (Radford et al., 2021) and LiT (Zhai et al., 2022): if a gap does exist, future work should work to address it. In the meanwhile, we recommend that future foundation models are benchmarked on significantly broader sets of downstream tasks, ideally with some analysis of the semantic distance between test and training data (if any). Alternatively, our results may be interpreted in light of the strong performance of uni-modal models. There may be a limited benefit of training from scratch on less curated multi-modal data sets compared to standard and well established uni-modal data sets, although we posit that at even larger scales the difference may be more significant.

**Uncertainty.** Another intriguing consequence of an explicit memory is that it enables trivial uncertainty estimations. If none of the $k$ nonzero dimensions of the relative representation of a new image is used to represent any label, all similarities would be null (or smaller than some threshold) and the model could be instructed to output an *unknown* token. We posit that this property is not specific to ASIF and that it could be generically implemented by many memory augmented networks, providing new interesting perspectives on the problem of estimating the prediction confidence on new, perhaps out-of-distribution, data.

**Limitations.** The biggest limitation of ASIF is the inference cost when scaling to the largest data sets. Additionally, while the simple ASIF procedure presented here offers a strong baseline for multimodal models, its performance still falls apart to CLIP and LiT when the mutimodal dataset is abundant and the cost of training is not a concern. At the same time we recognize that the experiments reported here are far from broadly covering the multitude of downstream tasks multimodal models are recognized to be useful to solve. Our scope here was to present the ASIF procedure and provide evidence of its effectiveness on the representative task of zero-shot classification that are common in the literature. In making this choice we followed (Zhai et al., 2022), that used the very same datasets to showcase the benefits of a locked image encoder, that is their main claim.

**Conclusions.** We presented a simple procedure called ASIF to assemble a fully functional multimodal model like CLIP from two unimodal pretrained encoders and a collection of image-text pairs without tuning a single neuron. While achieving competitive zero-shot classification results with its predecessors using a fraction of the data, the proposed ASIF procedure enriches multimodal models with editability –a new model based on different pairs can be deployed in seconds– and interpretable representations. The effectiveness of the ASIF procedure also clashes with the dominant view of a learning algorithm as a way to distill data into the parameters of a model, and raises questions on the role of memory and retrieval in machine learning.

## Reproducibility Statement

We describe in detail the ASIF procedure in the gray frame in Section 2. All the encoders and the datasets used in our experiments are publicly available. We have reported all the hyperparameters to fully reproduce the results in Table 1, the selection procedure, and the trends in Figure 4.

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
