# OpenReview forum: "ASIF: coupled data turns unimodal models to multimodal without training"
_ICLR.cc/2023/Conference — Submitted to ICLR 2023_

### Official Review · Reviewer_MKVK · 2022-10-24

**Confidence:** 3
**Correctness:** 2
**Technical Novelty And Significance:** 3
**Empirical Novelty And Significance:** Not applicable
**Recommendation:** 5

**Clarity, Quality, Novelty And Reproducibility:**

The paper is easy to read, and the proposed method can be easily implemented just by following the paper's instructions. However, one of the key contributions of the prosed method, data-efficiency, is not well-supported. The method has some technical novelty.

**Details Of Ethics Concerns:**

Not applicable: The paper does not have any ethical considerations to address.

**Strength And Weaknesses:**

Strength
- The paper is well-written and throughly addresses related work and implications of the proposed method.
- The proposed method is simple yet quite effective for multimodal zero-shot tasks using a relatively small paired dataset.

Weaknesses
- However, it seems that the model performance heavily depends on pretrained unimodal models, and the results reported in the paper come from the unimodal models that are trained on large-scale unimodal data: ImageNet21K (about 14M images) and 1B sentences scraped from the internet. I don't think that it is appropriate to say that ASIF is data efficient using such models. This seems to be more than the LiT training data. Do you have additional results using unimodal models that are trained on less data, such as ImageNet1K?
- Since the model performance also heavily depends on the ground-truth image-text pairs, you need to provide quantitative results on out-of-distribution tasks. I am not sure which dataset among ImageNet (or ImageNet v2), CIFAR100 and Pets is far distant in distribution to the subset of CC12M that you use for computing relative representations.

**Summary Of The Paper:**

The paper proposes a multimodal model that does not require any training on image-text pairs. The proposed non-parametric model, ASIF, leverages independent pretrained unimodal models to extract embeddings of data points in ground-truth image-text pairs. At inference, ASIF first computes the relative representation of a given input from one modality as the vector containing similarities between the given input and each data point from the dataset. It then treats this vector as the relative representation of the input's ideal correspondence from the other modality, and retrieves the candidate that has the most similar relative representation to the vector. With some sparsification treatments, ASIF achieves quite competitive zero-shot results on several image classification benchmarks, including ImageNet, ImageNet v2, CIFAR100 and Pets, using a relatively small paired dataset.

**Summary Of The Review:**

The proposed method is simple yet quite effective for building a multimodal non-parametric model, but one of the key contributions, data efficiency, should be validated with unimodal models that are trained on small-scale datasets.

---

> ### Author Response · Authors · 2022-11-19
> **Thanks for the kind review.**
>
> Thanks for the kind review.
>
> In light of the following discussion and the updated paper, we hope the reviewer will acknowledge our revision and upgrade the recommendation to full acceptance.
>
> - **About data-efficiency.** We are fully transparent about the need of large-scale unimodal dataset during the pretraining. From Section 3: *We recall that building an ASIF model from scratch still requires a lot of unimodal data to pretrain the encoders, but now this data may come untied and even without any label.*
> Our claim of data-efficiency is in line with the data-efficiency claim of LiT; we refer to the image-text pairs seen respect to previous methods (Fig. 1, Sec. 1, and 5.1 of the LiT paper).
>
> Following the reviewer suggestion, we report here and in the Appendix the results of ASIF based on DEIT [1], a visual encoder pretrained on **Imagenet1k**, where we observed similar performance (if not better) using the smaller pre-training dataset:
>
> |   ASIF Backbones and pre-training data                                                   | ImageNet | CIFAR | PETS | Imagenetv2 |
> |------------------------------------------------------|----------|-------|------|------------|
> | DEITbase (Im1k) - SentenceT                          | 60.9     | 50.2  | 81.5 | 52.2       |
> | (from Table1 in the paper) VIT (Im21k) -   SentenceT | 55.4     | 63.3  | 71.5 | 45.6       |
>
>
>
>
> - **About out-of-distribution tasks.** We remark that our choice of pre-training data (imagenet 21k), multi-modal (CC12M) and test datasets is identical to LiT (ASIF Table 1 and LiT Table 2). The CLIP training dataset is even private. The only additional data we use is for the pre-training of the language model, which we don’t fine-tune. Generally, we agree with the reviewer concern, and have dedicated a whole paragraph to discuss it: *Generalization to new distributions* paragraph in Section 4. However, we stress that the comparison against LiT is fair and follows the community standards.
> We’d like to remark that the strength of our approach lies in the fact that good open source unimodal models are widely available and we did not have to train these. Note that the most performing CLIP and LiT versions are **not** publicly available. ASIF significantly lowers the bar to obtain functionally equivalent models with comparable performance.
> We added the number of parameters and the pre-training dataset of backbone models in Table 2.
>
>
> [1] Touvron, H., et al. “Training data-efficient image transformers & distillation through attention,”. ICML 2021

---

> > ### Comment · Reviewer_MKVK · 2022-12-01
> > **Post-rebuttal review.**
> >
> > I would like to thank the authors for the rebuttal. The authors addressed all of my questions, especially in terms of the data efficiency. However, I am still not fully convinced that ASIF could be used as a baseline for foundation models such as CLIP and LiT. More specifically, the model performance deeply depends on the pretraining datasets; as shown in Table 2, DEIT trained on ImageNet1K outperforms the model trained on ImageNet21K in the tasks of ImageNet and ImageNetv2 by a substantial margin while they achieve opposite results on CIFAR100 and Pets. Unless the authors provide more consistent results by using larger and more diverse datasets, I am still leaning toward rejection.

---

> > > ### Author Response · Authors · 2022-12-05
> > > **The same is true for LiT**
> > >
> > > It is clear that the effectiveness of the ASIF method is influenced by the quality of the pretraining data. The same is true for the LiT approach, as demonstrated in Table 3 of the LiT paper. In that table, we can see that the Visual encoder, when pretrained only on ImageNet, outperforms the same encoder when it is pretrained on Imagenet21k in zero-shot classification on the Imagenet-test dataset. For the convenience of the reviewer, we have uploaded and highlighted the table to Dropbox (https://www.dropbox.com/sh/cguka8zoczi0xpp/AAAmOjkIZUOo_hdN5gz9nwCFa). The results of the DEIT experiment on ASIF align with these findings, as pretraining on Imagenet21k leads to better generalization on more diverse datasets like CIFAR or Pets, but can reduce performance on Imagenet-like datasets.
> > >
> > > Our goal is to demonstrate that it is possible to create multimodal models without the need for training a large neural network on a multimodal dataset. This could be a major advancement in the AI community, as it challenges the widely accepted notion that these abilities can only be obtained through the training of a large neural network on a multimodal dataset.
> > >
> > > If the reviewer believes that our findings are worth considering, we would be grateful if they could update the vote to a full accept.

---

### Official Review · Reviewer_nLjp · 2022-10-26

**Confidence:** 4
**Correctness:** 3
**Technical Novelty And Significance:** 3
**Empirical Novelty And Significance:** 3
**Recommendation:** 5

**Clarity, Quality, Novelty And Reproducibility:**

The paper is overall well-written. However, some important experiments mentioned in the Weakness section are expected to further validate the claims and support their conclusions.

**Strength And Weaknesses:**

## Strength
1. interesting idea for turning two unimodal pre-trained model into a multi-modal model w/o training.
1. the paper is well-written and easy to follow

## Weakness

1. The major assumption of the proposed method is that "captions of images that are close in the visual space to be themselves close in the language space." Please quantitatively prove this claim.

1. How does ASIF scale with respect to the size of multimodal dataset? In Figure 4, it looks like ASIF still has room for performance growth if more data are given. The reviewer is interested in seeing when the performance saturates. As the multimodal dataset scales up to 400M, what's the performance gap between ASIF and CLIP/LiT?

1. What's the relationship between the "quality" of the visual encoder and ASIF's performance? For example, do we need to take a larger model (in terms of #params) or a model trained with more data for ASIF to achieve better performance? Or maybe the pre-trained visual encoder's performance on eg ImageNet better correlates with ASIF's performance.

1. sensitivity of hyper-params p and k.
In Figure 4, it looks like the performance is quite sensitive to p and k. It would be good to show the grid search results with respect to these two parameters. Further, how sensitive are p and k to different target datasets? For example, the best (p, k) pair for ImageNet vs for Pets.

1. Even though the idea is interesting, what's the future of this method? According to the performance, it doesn't seem like a good alternative to existing multi-modal models such as CLIP or LiT.

1. In Table 1, it would be good to add a column of #params for each model.

**Summary Of The Paper:**

This paper proposed an interesting idea of turning two unimodal pre-trained models into a multi-modal model without training. Specifically, each sample is encoded into a relative representation with respect to paired multi-modal data in its own modality. The relative representation serves as the bridge between two modalities. ASIF achieves a non-trivial performance but still lags behind related works such as LIT and CLIP.

**Summary Of The Review:**

This paper proposed an interesting idea to bridge two uni-modal pre-trained models w/o training. However, some experiments are missing as detailed in the Weakness section.

---

> ### Author Response · Authors · 2022-11-19
> **Thanks for the kind review.**
>
> Thanks for the kind review.
>
> We have addressed all the points raised by the reviewer with new evidence, reported here and in the paper's appendix. We now address the key question:
>
> What is the future of the ASIF method?
> - ASIF needs no training, it makes building a multimodal model radically more affordable. Researchers with limited computing resources may find it very interesting for applications where CLIP capabilities are needed but training the large neural encoders from scratch is prohibitive.
> - With ASIF, it is trivial to adjust a multimodal model by adding or forgetting specific samples. The latter use-case seems to have a quite bright future, as the right to use specific assets often change over time and removing the effect of specific samples from a trained network today requires sophisticated forgetting techniques.
> - ASIF offers competitive performance already with very small image-text datasets, a feature that could broaden the adoption of image-text models in the future.
> - In image classification, it is possible to trace back each ASIF prediction to a small set of entries in the training dataset, making it easily interpretable. This property could well justify adopting an ASIF model rather than a CLIP or LIT one for sensitive tasks in the future.
>
> The other weaknesses:
> 1. Captions of similar images are themselves similar. We now have a quantitative proof. Please look at Figure 6 in the Appendix.
> 2. We share the same curiosity as the reviewer, but were not able to scale to a larger multimodal dataset due to limited computing resources. However, we tried smaller encoders to see if they saturated earlier, see Figure 8. We found that the accuracy keeps growing without saturating but is lower for smaller models.
> 3. Larger visual encoders correlate with better ASIF performance, we tried three versions of DEIT [1] different in size. DEIT models are trained on Imagenet1k and outperform the original VIT [2] trained on Imagenet21k on all test datasets except CIFAR. Please look at Figure 8 and Table 2 in the Appendix.
> 4. The performance is not very sensitive to p and k. Please look at the grid searches in Figure 9 in the Appendix.
>
> In light of the above discussion and the updated paper, we hope the reviewer will acknowledge our revision and upgrade the recommendation to full acceptance.
>
> [1] Touvron, Hugo, et al. "Training data-efficient image transformers & distillation through attention." International Conference on Machine Learning. PMLR, 2021.
>
> [2] Dosovitskiy, Alexey, et al. "An Image is Worth 16x16 Words: Transformers for Image Recognition at Scale." International Conference on Learning Representations. 2020.

---

### Official Review · Reviewer_hNdq · 2022-10-26

**Confidence:** 3
**Clarity, Quality, Novelty And Reproducibility:** The paper is clear and easy to follow…
**Correctness:** 3
**Technical Novelty And Significance:** 2
**Empirical Novelty And Significance:** 3
**Recommendation:** 5

**Strength And Weaknesses:**

Pro: It it interesting to see such an effort on tuning unimodal encoders into multimodal models using non-parametric methods.

---

Con: clarification needed on relationship with respect to K Nearest Neighbor.

As noted by the authors, ASIF, in essence, is “making predictions on new samples by exploiting the similarity with a dictionary of previous data points”. This makes ASIF extremely similar to KNN. The first half of ASIP, i.e., finding the k most-similar images, looks the same as KNN. What differs is how to find the best caption given the k “neighbor” image-caption pairs. I would imagine a naive approach of finding the caption from the training set that is mostly similar to k captions (using Language Model similarity) would do pretty well (similar to the majority voting in KNN). The author also provides a nice discussion in Sec 4 “learning or retrieval”. However, if we link ASIF to KNN, then it would be clear that ASIF, like KNN, counts as a non-parametric supervised learning algorithm.

In fact, KNN has been used for evaluating self-supervised visual representation models (e.g., Table 1 of Zhou et al.) and the performance is already high (78.0). But I can see that the performance is not directly comparable as Zhou et al. used ImageNet as the KNN training set while this paper uses Conceptual Captions (CC). The impressive results in the paper might look less “surprising”, if we were to believe that ImageNet and CC have a lot of overlap. Then the current results would be somewhat a domain-shifted version of the KNN experiments in Zhou et al.

I would thus appreciate a discussion on:

 1) what makes ASIF different from and potentially better than KNN (and similar methods);

 2) experiments on a naive KNN baseline on using CC data for ImageNet.

Zhou et al., iBOT: Image BERT Pre-Training with Online Tokenizer

---

Minor questions:

For the image classification experiments, ASIF will select a caption from the 1.6M captions. How should we map the retrieved caption to a label? Is it a simple rule-based ngram match?


**Summary Of The Paper:**

The paper proposes ASIF, which can retrieve relevant captions given an image **without training**, utilizing 1) two pre-trained unimodal encoders and 2) a relatively small multimodal datasets. The key intuitive is that “captions of similar images should be themselves similar”; thus the retrieval is done through the relative distances given by the two pre-trained unimodal encoders.

In experiments, the authors show decent performance on several image classification datasets, using conceptual captions as the multi-modal dataset.

**Summary Of The Review:**

I very much like the intuition of ASIF. It works decently in practice and has several nice properties nicely argued by the authors.

However, my main concern is that the method is very similar to KNN and once we consider it as some kind of KNN, the analysis and results look less “surprising”.

---

> ### Author Response · Authors · 2022-11-19
> **Thanks for the helpful review.**
>
> Thanks for the helpful review.
>
> We address the main questions: *For the image classification experiments, ASIF will select a caption from the 1.6M captions. How should we map the retrieved caption to a label? How is this different than KNN*
>
> - **ASIF will not select a caption from the 1.6M captions and is functionally equivalent to CLIP**: ASIF is performing open vocabulary classification. It takes as input an image and a *free form text*, and outputs a scalar $\in [-1;1]$ representing the coherence between the image and the input text, enabling *Zero-shot* generalization to new tasks described by the captions.
> For example, when we do image classification on PETS, we craft a set of captions for each of the 37 classes using templates ('a type of pet {CLASS}.',  'a {CLASS} texture.', '{CLASS} , an animal.'), so ASIF selects one from 37$\times$3 = 111 captions. When we classify Imagenet, we have 7 templates, so ASIF selects one from 7$\times$1000 captions. We can measure similarity with unseen captions in the text space only through relative representations (Figure 2).
>
> - **Difference with iBot**:  iBot takes a set of labeled images at test time *from the same distribution* and measures similarity against those labels through KNN. This is *few-shot* classification. Here, we generalize *zero-shot* to new labels. For example, Zero-shot classification shown in Figure 3 with two competing original captions can be solved only by CLIP-like models like ASIF, not by iBot-like models with KNN.
>
> - **Interpretation of ASIF as a KNN:** We agree with the reivewer that this connection needed strengthening, we have now added a discussion in a new paragraph just before subsection 2.1. Note that there is no overlap in the data sets we use and that our experimental evaluation is standard in the community, it is identical to LiT in both pre-training, multi-modal, and test data, so the generalization results are highly non-trivial.

---

### Decision · Program_Chairs · 2023-01-20

**Decision:**

Reject

**Justification For Why Not Higher Score:**

It is unclear in which scenarios the proposed method is actually useful.

**Justification For Why Not Lower Score:**

N/A

**Metareview: Summary, Strengths And Weaknesses:**

The paper studies an approach that creates a CLIP-like model from a pre-trained image encoder and a pre-trained text encoder using a limited number of image-text pairs using a non-parametric approach. The approach concatenates both encodings and performs nearest neighbor prediction using a dissimilarity-based representation of the resulting representations.

The reviewers are lukewarm on the paper. One of the main concerns they raise is that it is unclear in what scenarios this would be a useful alternative to the current state-of-the-art. The proposed approach achieves acceptable results using 1.6M image-text pairs, but the performs is substantially worse than that of CLIP / LIT (including variants of LIT that are trained on relatively small datasets of publicly available image-text pairs). If such data is publicly available, why wouldn’t one use it?

It should also be noted that the proposed approach is a very standard non-parametric approach that does not scale particularly well, presuming that k needs to increase logarithmically with the training set size for optimal results (as the results in Figure 4 suggest).